# Treadmill Exercise after Controlled Abnormal Joint Movement Inhibits Cartilage Degeneration and Synovitis

**DOI:** 10.3390/life11040303

**Published:** 2021-04-01

**Authors:** Yuichiro Oka, Kenij Murata, Kaichi Ozone, Takuma Kano, Yuki Minegishi, Aya Kuro-Nakajima, Kohei Arakawa, Takanori Kokubun, Naohiko Kanemura

**Affiliations:** 1Department of Health and Social Services, Health and Social Services, Graduate School of Saitama Prefectural University, Koshigaya, Saitama 343-8540, Japan; 2191001s@spu.ac.jp (Y.O.); 2191003x@spu.ac.jp (K.O.); 2291004n@spu.ac.jp (T.K.); 2191004f@spu.ac.jp (Y.M.); 2181301f@spu.ac.jp (A.K.-N.); murata-kenji@spu.ac.jp (K.A.); 2Department of Physical Therapy, Health and Social Services, Saitama Prefectural University, Koshigaya, Saitama 343-8540, Japan; 2191002a@spu.ac.jp (K.M.); kokubun-takanori@spu.ac.jp (T.K.)

**Keywords:** anterior cruciate ligament, cartilage degeneration, controlled abnormal joint movement, osteoarthritis, treadmill exercise

## Abstract

Cartilage degeneration is the main pathological component of knee osteoarthritis (OA), but no effective treatment for its control exists. Although exercise can inhibit OA, the abnormal joint movement with knee OA must be managed to perform exercise. Our aims were to determine how controlling abnormal joint movement and treadmill exercise can suppress cartilage degeneration, to analyze the tissues surrounding articular cartilage, and to clarify the effect of treatment. Twelve-week-old ICR mice (*n* = 24) underwent anterior cruciate ligament transection (ACL-T) surgery on their right knees and were divided into three groups as follows: ACL-T, animals in the walking group subjected to ACL-T; controlled abnormal joint movement (CAJM), and CAJM with exercise (CAJM + Ex) (*n* = 8/group). Walking-group animals were subjected to treadmill exercise 6 weeks after surgery, including walking for 18 m/min, 30 min/day, 3 days/week for 8 weeks. Safranin-O staining, hematoxylin-eosin staining, and immunohistochemical staining were performed. The OARSI (Osteoarthritis research Society international) score was lower in the CAJM group than in the ACL-T group and was even lower in the CAJM + Ex group. The CAJM group had a lower meniscal injury score than the ACL-T group, and the CAJM + Ex group demonstrated a less severe synovitis than the ACL-T and CAJM groups. The observed difference in the perichondrium tissue damage score depending on the intervention method suggests different therapeutic effects, that normalizing joint motion can solve local problems in the knee joint, and that the anti-inflammatory effect of treadmill exercise can suppress cartilage degeneration.

## 1. Introduction

Knee osteoarthritis (OA) is one of the most common musculoskeletal diseases, characterized by degeneration of the articular cartilage. It is a disease that develops due to the complex involvement of many factors such as aging, obesity, genetics, and mechanical stress [1,2,3,4,5], and its pathogenesis remains unclear. Currently, medication, physiotherapy, such as LLLT [6,7,8,9,10], and physical therapy, are the main conservative treatments for knee OA. However, although these therapies have been reported to be effective for reducing pain [11], which is the main complaint associated with knee OA, the establishment of disease-modifying therapies that inhibit cartilage degeneration has not been realized. With the progression of cartilage degeneration, knee OA is associated with pathological changes such as synovitis, sub-chondral bone lesions, and osteophytes, resulting in increased pain and decreased motor function. Surgical treatments such as UKA (unicompartmental knee Arthroplasty) and TKA (total knee arthroplasty) are needed when the lesions are advanced, and the establishment of a treatment method is essential to improve the quality of life for patients with OA and to reduce the burden on society.

In recent years, there have been reports on the effects of exercise therapy using animal OA models, and some studies have aimed to suppress cartilage degeneration via physical exercise such as treadmill exercise. Moderate mechanical stress is necessary to maintain cartilage homeostasis, and it is thought that OA develops and progresses when the balance is disturbed. Previous studies have reported that exercise increases extra-cellular matrix synthesis and suppresses inflammatory factors [12], and moderate-intensity treadmill exercise in OA model rats was found to suppress cartilage degeneration more than that in the non-exercise group [13,14,15,16,17,18,19,20]. In other words, physical exercise such as treadmill exercise could be a part of the treatment for OA.

The anterior cruciate ligament transection (ACL-T) model, which is a commonly used animal model of surgically induced OA, involves cutting the ACL to induce joint instability and progression to OA. In contrast, the controlled abnormal joint movement (CAJM) model, proposed by Murata et al. [21,22] and Onitsuka et al. [23], involves ACL-T followed by another surgical procedure to re-stabilize the joint and reduce joint instability. In the CAJM model, the expression of inflammatory factors and cartilage matrix-degrading factors is suppressed, joint instability is reduced, and progression to OA is slower. Previous studies often used models involving ACL-T or destabilization of the medial meniscus to examine the effect of exercise intervention. However, in all such studies, exercise intervention was performed in the presence of joint instability caused by ligament rupture. We found that the state of the intra-articular environment can greatly influence the effect of exercise on cartilage degeneration, even if exercise is performed under the same conditions [24].

We showed that treadmill exercise after improving joint motion further suppresses cartilage degeneration. However, the mechanism by which cartilage degeneration is suppressed is still unclear. Mechanical stress is a factor for these therapeutic effects. It has been reported that controlling joint movement reduces mechanical stress and suppresses cartilage degeneration and osteophyte maturation, and we believe that the anti-inflammatory effects of exercise are realized in this environment. In recent years, OA has been regarded as a whole joint disease, including synovitis, subchondral bone lesions, and abnormal function of the meniscus, and the suppression of lesions in the tissues surrounding the articular cartilage might help to prevent cartilage degeneration. In other words, it is thought that the positive effects of controlling joint movement and treadmill exercise on the tissues around the cartilage will lead to the suppression of cartilage de-generation. The purpose of this study was to assess the lesions in the tissues surrounding the articular cartilage and to clarify the effects of controlling joint movement and treadmill exercise on this disease. 

## 2. Materials and Methods

### 2.1. Research Design

This study was approved by the Animal Research Committee of Saitama Prefectural University (approval number: 29-12), and the animals were handled in accordance with the relevant legislation and institutional guidelines for humane animal treatment. Twelve-week-old male ICR mice (*n* = 24) from Japan SLC Inc. (Shizuoka, Japan) were used in this study. The mice were divided into three groups (*n* = 8/group) as follows: ACL-T group (untreated after ACL-T), controlled abnormal joint movement (CAJM) group (controlling joint movement only), and CAJM + Ex group (controlling joint movement followed by treadmill exercise) (Figure 1). Mice were housed in cages at two mice per cage and a room temperature of 23 ± 1 °C and relative humidity of 55 ± 5%. The day/night cycle was set at 12 h, feed and water were provided ad libitum, and exercise was not restricted. 

### 2.2. Surgical Procedure

The animals were anesthetized via the inhalation of diethyl ether (Nacalai Tesque Co., Ltd., Kyoto, Japan), followed by the administration of a triple mixture of anesthetics (Domitor, 10 mL (Nippon Zenyaku Kohgyo Co., Ltd., Fukushima, Japan); dolmicam injection, 10 mg (Astellas Pharma Inc, Tokyo, Japan); medetomidine antagonists (Antisedan^®^, 10 mL; Nippon Zenyaku Kohgyo Co., Ltd. Japan and Betolphal^®^, 5 mg (Meiji Seika Pharma Co., Ltd. Tokyo, Japan) and saline in the proportions of 1.875 mL, 2 mL, 2.5 mL, and 18.625 mL, respectively, in a 25 mL preparation), and saline at a ratio of 0.15 mL and 9.85 mL, respectively, in a 10 mL volume, which was administered subcutaneously at a dose of 0.1 mL per 10 g of body weight for deep an-esthesia and pain relief. An incision was made in the anterior aspect of the knee joint to expose the joint capsule, and then, the ACL was cut by penetrating the capsule with a shear blade from the medial side of the patellar tendon. In the CAJM model, the femur and tibia were perforated and looped with 3-0 nylon thread to control the anterior instability of the tibia [24]. In the ACL-T group, the instability of the tibia was not limited by loosely applying nylon thread loops.

### 2.3. Exercise Intervention

The CAJM + Ex group was subjected to physical exercise using a rodent treadmill. All mice were allowed to become familiarized with the treadmill environment for 3 days. Interventions were performed 3 days per week for 8 weeks according to our protocol, with a daily intervention time of 30 min at a speed of 18 m/min.

### 2.4. Evaluation of Tibial Malposition

To evaluate joint instability, photographs were taken using a soft X-ray system. The right hind leg was photographed in the 90° flexion position of the knee joint after pruning the muscles. The imaging conditions were a voltage of 28 kV, current of 1.5 mA, and exposure time of 1 s. The images were digitized using a digital image sensor NAOMI (RF Co., Ltd. Nagano, Japan). The joint instability of the knee joint was quantified as the amount of anterior translation of the tibia with the image processing software Image J (im-agej.nih.gov).

### 2.5. Histological Analysis

Histological analysis was performed to evaluate the lesions of the articular cartilage, subchondral bone, meniscus, and synovium. The collected right knee joints were washed with saline and fixed in 4% PFA for 24 h. They were then demineralized with 10% EDTA for 21 days. After demineralization, the samples were dehydrated with ethanol and replaced with xylene. Paraffin blocks were prepared using the paraffin dispenser TEC-P-DC-J0. Subsequently, sagittal sections (7 μm) were prepared using a microtome ROM-380 (Yamato Kohki Industrial Co., Ltd., Saitama, Japan).

To evaluate the articular cartilage, subchondral bone, and meniscus, Safranin O Fast Green staining was performed according to the method of Schmitz et al. The stained sections were photographed using an all-in-one fluorescence microscope BZ-X700 (KEYENCE Co., Osaka, Japan). Articular cartilage was evaluated based on the scoring method recommended by OARSI [25]. The posterior meniscus was evaluated with reference to the report by Kwok et al. [26]. The total scores for all criteria (structural, cellularity, and matrix staining) ranged from 0 to 21. The subchondral bone was analyzed with reference to the report by Aho et al. [27]. HE (hematoxylin–eosin) staining was performed to evaluate the synovitis of the patellar groove. The evaluation of synovitis was performed based on the OARSI histopathology initiative [28] and by measuring synovial thickness. The criteria for the synovial inflammation score are based on increased numbers of synovial lining cell layers, the proliferation of subsynovial tissue, and infiltration of inflammatory cells. Synovial thickness was quantified using dedicated image analysis software (Image J; National Institutes of Health, Bethesda, MD, USA). Moreover The synovial thickness was measured at three points per section. Since the evaluation was based on two sections, the average of the six points was used as the representative value. Two independent observers performed scoring, and the average value was retained. For the evaluation of cartilage, meniscus, subchondral bone, and synovitis, two sections were evaluated each. The two sections were spaced 100 μm apart. 

### 2.6. Immunohistochemistry (IHC) Analysis

Immunostaining was performed to evaluate the activity of factors involved in cartilage degeneration and synovitis. IHC was performed using the Avidin-Biotinylated enzyme complex method based on the protocol of the VECTASTAIN Elite ABC Rabbit IgG Kit (Vector Laboratories, CA, USA). Tissue sections were deparaffinized with xylene and ethanol and then immersed in ethanol containing 0.3% hydrogen peroxide (FUJIFILM Wako Pure Chemical Co., Osaka, Japan) for 30 min to deplete endogenous enzyme activity. Non-specific binding of primary antibodies was blocked by immersion in normal animal serum (Vector Laboratories, USA) for 20 min, and then, anti-MMP13 antibody (ab39012; Abcam plc., Japan, dilution 1/250), anti-Gremlin-1 antibody (ab Japan, diluted concentration 1/200), and anti-TNF-α antibody (BS-2081R; Bioss plc. diluted 1/100) were applied. The reaction was carried out for more than 8 h. For the secondary antibody, anti-rabbit IgG Biotinylated Antibody (Vector Laboratories, USA) was used, and the reaction was carried out for 30 min. As a sensitizing reaction, VECTASTAIN ABC Rabbit IgG Kit (Vector Laboratories, USA) was used for 30 min. Finally, color was generated by diaminobenzidine (Dako Japan Co., Ltd., Japan). Gremlin-1 and MMP13 were analyzed in the articular cartilage; TNF-α was analyzed in the synovium and infrapatellar fat pad (IFP). For analysis, the percentage of positive cells was calculated as the number of positive cell relative to the cell containing nuclei, stained by hematoxylin, in a randomly selected articular cartilage area of 10000 (100 × 100) μm^2^.

### 2.7. Statistical Analysis

All data were tested for normality by the Shapiro-Wilk test. The tibia anterior displacement data were compared among the three experimental groups (ACL-T, CAJM, CAJM + Ex) using analysis of variance with Tukey’s post hoc test. The OARSI score, meniscus score, subchondral bone score, synovium inflammation score, the percentages of Gremlin-1-, MMP-13-, and TNF-α-positive cells were compared among groups using the Kruskal-Wallis test with a Steel-Dwass post hoc analysis. Parametric data are shown as mean values [95% confidence intervals], and non-parametric data are shown as median values [interquartile range]. All significance levels were set at <5%.

## 3. Results

### 3.1. CAJM Suppresses Anterior Displacement of the Tibia

After tissue collection, the knee joint was photographed in the sagittal plane using soft X-rays, and the anterior deviation of the tibia was measured (Figure 2). The amount of anterior displacement was 0.84 [0.60–1.07] mm in the ACL-T group, 0.45 [0.30–0.60] mm in the CAJM group, and 0.57 [0.47–0.67] mm in the CAJM + Ex group. The CAJM and CAJM + Ex groups exhibited significantly decreased anterior displacement compared to that in the ACL-T group ([ACL-T vs. CAJM] *p* = 0.031, [ACL-T vs. CAJM + Ex] *p* = 0.007), and there was no significant difference in anterior displacement between the CAJM and CAJM + Ex groups. 

### 3.2. Treadmill Exercise After CAJM Inhibits Cartilage Degeneration 

Histological images and scoring results of Safranin-O Fast Green staining are shown in Figure 3A. In the ACL-T group, the cartilage loss reached the middle layer, whereas in the CAJM group, cartilage loss was limited to the superficial layer, and in the CAJM + Ex group, the cartilage loss was limited to superficial cracks and decreased staining was observed. The CAJM group had a lower OARSI score than the ACL-T group, and the CAJM + Ex group had even less cartilage degeneration ([ACL-T vs. CAJM] *p* = 0.048, [ACL-T vs. CAJM + Ex] *p* = 0.013, [CAJM vs. CAJM + Ex] *p* = 0.042) (ACL-T: 3.0 [3.0–4.0], CAJM: 2.5 [2.0–3.0], CAJM + Ex: 2 [1.75–2.0]) (Figure 3B). 

Regarding the meniscus injury score, the CAJM group had significantly reduced scores compared to those in the ACL-T group, and there was no significant difference between the CAJM and CAJM + Ex groups ([ACL-T vs. CAJM] *p* = 0.039) (ACL-T: 14.5 [12.8–16.3], CAJM: 8.0 [5.8–9.3], CAJM + Ex: 9.0 [8.0–11.3]) (Figure 4A,B). There was also no significant difference in subchondral bone damage scores among the three groups (*p* = 0.893; ACL-T: 2.0 [1.0–2.0], CAJM: 2.0 [1.0–2.0], CAJM + Ex: 2.0 [1.8–2.3]) (Figure 4A,C). 

### 3.3. CAJM Inhibits the Expression of Gremlin-1 

Histological images obtained by immunohistochemical staining showed cells positive for Gremlin-1 in the superficial and deep layers of articular cartilage, and some positive findings were also observed in the extracellular matrix of the deep cartilage layers (Figure 5A). In terms of the positive cell rate, the CAJM group showed significantly lower values than the ACL-T group ([ACL-T vs. CAJM] *p* = 0.047) (ACL-T: 67.4 [58.5–74.3] %, CAJM: 57.0 [50.5–62.0] %, CAJM + Ex: 62.3 [55.3–73.3] %) (Figure 5B). For MMP-13, positive cells were observed throughout the entire articular cartilage (Figure 5A). In terms of positive cell rates, the CAJM and CAJM + Ex groups showed a lower value than the ACL-T group ([ACL-T vs. CAJM] *p* = 0.023, [ACL-T vs. CAJM + Ex] *p* = 0.025) (ACL-T: 74.4 [70.3–80.5] %, CAJM: 55.9 [49.8–66.0] %, CAJM + Ex: 56.5 [52.5–61.8] %) (Figure 5B). 

### 3.4. Treadmill Exercise Suppresses Synovitis

In terms of synovitis scores, the CAJM + Ex group showed lower inflammation scores than the ACL-T and CAJM groups ([ACL-T vs. CAJM + Ex] *p* = 0.003, [CAJM vs. CAJM + Ex] *p* = 0.040) (ACL-T: 4.0 [3.8–5.0], CAJM: 3.5 [3.0–4.0], CAJM + Ex: 2.0 [1.0–2.3]) (Figure 6C). In terms of synovial thickness, the CAJM + Ex group showed lower than the ACL-T and CAJM groups ([ACL-T vs. CAJM + Ex] *p* < 0.001, [CAJM vs. CAJM + Ex] *p* = 0.032) (ACL-T: 119.5 [96.0–142.5] μm, CAJM: 110.5 [91.8–120.5] μm, CAJM + Ex: 72.0 [63.5–76.0] μm) (Figure 6B). For the analysis in the synovium, the results of TNF-α are shown. Positively stained cells were observed in the synovium and IFP. The number of positive cells was significantly lower in the CAJM + Ex group than in the ACL-T and CAJM groups ([ACL-T vs. CAJM + Ex] *p* = 0.006, [CAJM vs. CAJM + Ex] *p* = 0.050) (ACL-T: 65.3 [57.3–73.5] %, CAJM: 65.6 [62.5–70.3] %, CAJM + Ex: 51.1 [47.5–56.0] %) (Figure 6D). 

## 4. Discussion 

The purpose of this study was to clarify the effects of interventions to manage abnormal joint motion and subsequent treadmill exercise in a mouse model of early-stage OA by evaluating not only the articular cartilage but also the surrounding tissues. In a previous study, we pointed out the need to consider the kinematics of the knee joint itself to perform treadmill exercise and reported that exercise with abnormal joint motion can lead to progressive OA changes. In response to this, this study attempted to elucidate why such interventions are necessary. We found that interventions focusing on the knee joint involved increasing or decreasing mechanical stress on the articular cartilage and that interventions with treadmill exercise might involve factors other than mechanical stress.

In the comparison between the ACL-T and CAJM groups, cartilage degeneration was suppressed in the CAJM group. Murata et al. reported a delay in the onset of OA compared to that with ACL-T at 8 weeks after CAJM in a rat model [21], which is similar to the results of previous studies. Moreover, Oka et al. reported that CAJM suppressed the expression of inflammatory and substrate-degrading factors in mice with OA, although there was no difference in cartilage degeneration at 4 weeks [24]. In this study, we examined the effect of intervention at 8 weeks in a mouse model of OA and found that cartilage degeneration was suppressed in the CAJM group through the downregulation of factors related to OA changes. In this study, we also analyzed Gremlin-1, which has been reported to be generated when excessive mechanical stress is added to chondrocytes and to cause OA via the NF-κB pathway [29]. In the present study, the positive cell rate of Gremlin-1 was lower in the CAJM group than in the ACL-T group, which might indicate that CAJM reduced mechanical stress on the cartilage. In the ACL-T model, ACL transection increases the anterior instability of the tibia and changes the contact area between the femur and tibia, resulting in the development and progression of OA in the posterior part of the cartilage [30]. The CAJM model, in contrast, is a model in which the function of the ACL is augmented by surgical treatment from outside the joint capsule to suppress the anterior instability of the tibia [31]. The soft X-ray results also confirmed the anterior deviation of the tibia in the ACL-T group, whereas the anterior deviation of the tibia was suppressed in the group that adopted the CAJM model. These data predict a difference in contact conditions between the ACL-T and CAJM models and support the rationale for accelerated posterior cartilage degeneration of the tibia in the ACL-T group. MMP-13 is a proteolytic enzyme and is deeply involved in the onset and progression of OA. It has been reported that MMP-13 is particularly expressed when the Gremlin-1 pathway is activated. In this study, MMP-13 was decreased in the CAJM group, suggesting that a decrease in mechanical stress is involved in the suppression of MMP-13 expression. Murata et al. reported that the CAJM model shows suppressed osteophyte maturation through the downregulation of TGFβ expression and inhibition of the activity of signaling that causes endochondral ossification [32]. It is believed that osteophytes are formed by increased mechanical stress, and the suppression of Gremlin-1 and the results of previous studies suggest that CAJM reduces mechanical stress on articular cartilage. Tissues related to mechanical stress on articular cartilage include the meniscus and subchondral bone. The meniscus and subchondral bone are adjacent to the articular cartilage and absorb shock during the load response [33,34]. In the present study, there was a difference in the damage score of the meniscus between the CAJM and ACL-T groups. In the ACL-T model, it is assumed that the contact area changes behind the articular cartilage due to ACL dysfunction. With CAJM, the contact environment was controlled to avoid concentrating mechanical stress on the meniscus, which might have led to the suppression of damage. This change in the function of the meniscus could have caused an increase or decrease in mechanical stress on the articular cartilage, leading to cartilage degeneration in the CAJM model.

However, although there was a difference in OARSI scores between the CAJM and CAJM + Ex groups, there was no difference in the meniscus or subchondral bone damage scores. This suggests that the mechanism of treadmill exercise as a chondroprotective effect is less related to the increase or decrease in mechanical stress. The fact that the ACL-T group had more advanced degeneration than the CAJM and CAJM + Ex groups, in terms of meniscal damage scores, indicates that the environment within the knee joint is more important than exercise in terms of effects on the meniscus. In the CAJM + Ex group, the stress added to the meniscus by treadmill exercise might have been increased more than that in the CAJM group. In other words, the results of our previous study showed that cartilage degeneration can be prevented if the environment in the joint is controlled. Furthermore, although there was a difference in the OARSI scores between the CAJM and CAJM + Ex groups, there was no difference in the positive cell rates of Gremlin-1 and MMP-13 in the articular cartilage analysis. Gremlin-1 is thought to be expressed in a stress-dependent manner, and the increased number of steps in the CAJM + Ex group, due to exercise, might be the reason for the lack of a difference between the CAJM and CAJM + Ex groups. In addition to MMP-13, many other factors have been reported to degrade the extracellular matrix of articular cartilage, and it is possible that other factors suppressed the OARSI score in the CAJM + Ex group, as compared to that in the CAJM group.

In addition, we performed histological analysis of the synovium to clarify the mechanism by which physical exercise inhibits cartilage degeneration. There was a difference in synovitis scores between the CAJM and CAJM + Ex groups. Furthermore, TNF-α, which indicates that inflammation is occurring, was also suppressed in the CAJM + Ex group, suggesting that the chondroprotective effect of exercise is mediated by the synovium. In recent years, OA has been recognized as an inflammatory disease, and the involvement of immune response mechanisms, including synovial-mediated inflammatory responses, has been attracting attention. The mechanism underlying synovitis includes cartilage debris and dead cells that are recognized as DAMPs, which activate M1 macrophages that produce inflammatory and substrate-degrading factors [35,36,37]. The factors produced by M1 macrophages accelerate cartilage degeneration. Indeed, previous studies have shown that besides synovial inflammation, infrapatellar fat pad inflammation is involved in the progression of OA [38]. It has also been reported that exercise suppresses inflammation in adipose tissue by reducing macrophage infiltration [39]. Although we did not identify the cells of the synovium and IFP in this study, exercise possibly suppressed inflammation of IFP by switching the phenotype of macrophages. Since synovitis was suppressed only in the CAJM + Ex group that exercised in the present study, treadmill exercise might be one therapeutic strategy to stop the cycle of synovitis and cartilage degeneration.

Previous reports have shown that moderate mechanical stress increases the pro-duction of cartilage matrix synthesis factors, such as TGF-β and Sox9, in chondrocytes in vitro [40,41], suggesting that the mechanism of chondroprotection mediated by exercise therapy is the addition of moderate mechanical stress. The results of the present study suggest that the chondroprotective effect of exercise therapy is due to the added mechanical stress. However, there was no difference in the damage to tissues adjacent to articular cartilage, such as subchondral bone and the meniscus. Furthermore, in animal OA models, cartilage degeneration is suppressed by treadmill exercise in an environment where excessive mechanical stress is added by ACL-T, making it difficult to understand the effect of exercise therapy based on mechanical stress alone. Aerobic exercise, such as treadmill exercise, is thought to have anti-inflammatory effects due to its systemic influence. It has been reported that aerobic exercise not only decreases inflammatory cytokines such as IFN-γ and TNF-α in blood monocytes but also increases anti-inflammatory factors such as IL-10 and IL-4 [42,43]. Furthermore, it has also been reported that exercise decreases inflammatory monocytes. This response might indicate that exercise improves the inflammatory state in the blood, decreases inflammatory macrophages, and increases anti-inflammatory macrophages in the synovium, leading to the suppression of cartilage degeneration.

It is also possible that the suppression of synovitis was not caused by an improvement in the systemic inflammatory state. Since synovitis in OA is usually caused by cartilage damage, it is possible that synovitis occurred because cartilage degeneration was suppressed in this study. The analysis of the meniscus and subchondral bone also did not verify the type of reaction that occurred in those tissues. It is possible that molecular anabolic signaling pathways are activated by exercise in these tissues, and the results of this study alone cannot clarify what suppressed synovitis. However, the fact that synovitis was not suppressed by CAJM, which can be used to manage mechanical stress, indicates that synovitis was caused by exercise. Whether the mechanical addition of exercise had an effect on the knee joint itself or on the whole body must be verified in the future.

Another limitation of this study is that the function of the meniscus and subchondral bone was only examined in terms of structural changes in the tissue. It is difficult to explain the causal relationship only by comparing the histological damage scores. By examining the factors expressed in each tissue and additional time-series data, it will be possible to clarify the mechanism by which joint braking and treadmill exercise suppress cartilage degeneration.

## 5. Conclusions

The present study examined the effect of treadmill exercise on the suppression of abnormal joint motion in relation to the tissues surrounding the cartilage. The results showed that the suppression of abnormal joint motion might have inhibited damage to the meniscus and thus suppressed cartilage degeneration. We also found that treadmill exercise suppressed synovitis and that the anti-inflammatory effect of systemic exercise could be involved in the suppression of cartilage degeneration. Further clinical data will form the basis to clarify the therapeutic effect, which could contribute to the establishment of a treatment regimen for knee OA.

## Figures and Tables

**Figure 1 life-11-00303-f001:**
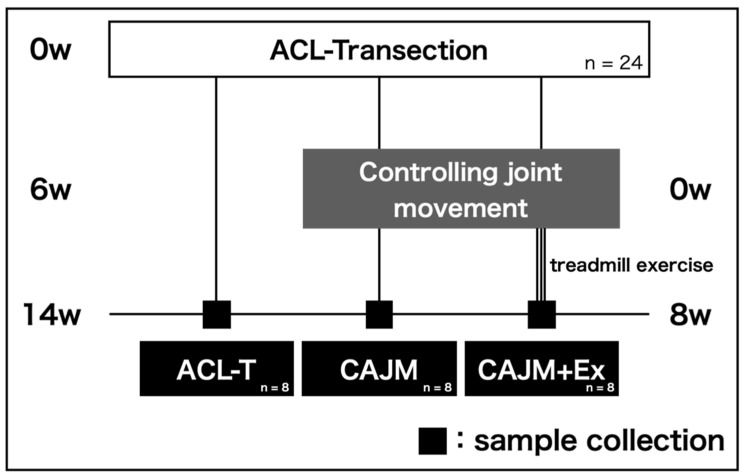
Protocol of X-ray radiography, histological evaluation, and immunohistochemical evaluation. These analyses were performed on the anterior cruciate ligament transection (ACL-T) group, the controlled abnormal joint movement (CAJM) group without exercise intervention, and the CAJM group with exercise intervention (CAJM + Ex) (each group, *n* = 8).

**Figure 2 life-11-00303-f002:**
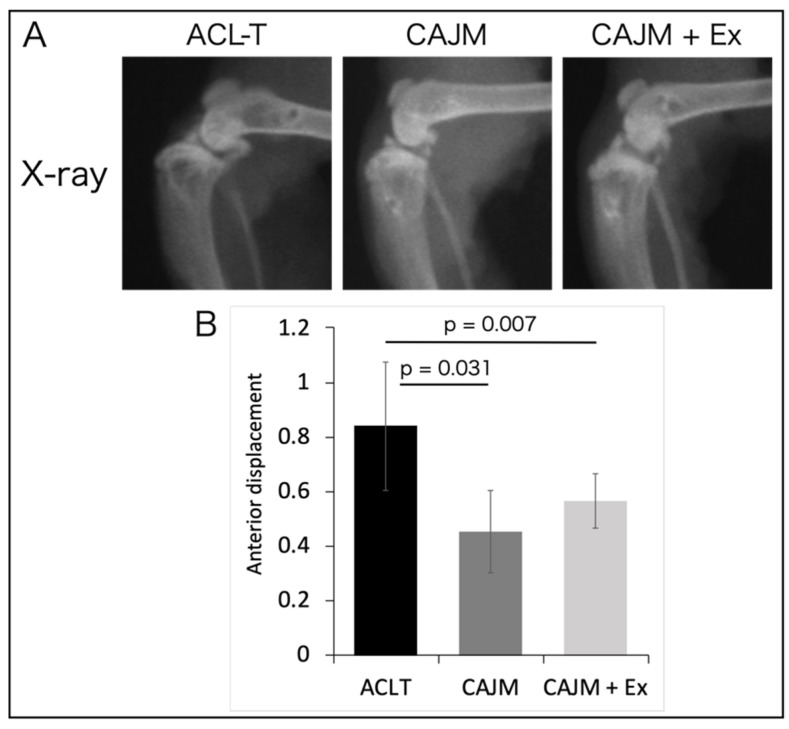
Soft X-ray radiographs of the knee joint (**A**) and measurement of the amount of anterior displacement of the tibia (**B**). The CAJM model exhibited suppressed anterior displacement of the tibia, and the stability of the knee joint was regained (each group, *n* = 8). ACL-T, anterior cruciate ligament transection group; CAJM, controlled abnormal joint movement; Ex, treadmill exercise.

**Figure 3 life-11-00303-f003:**
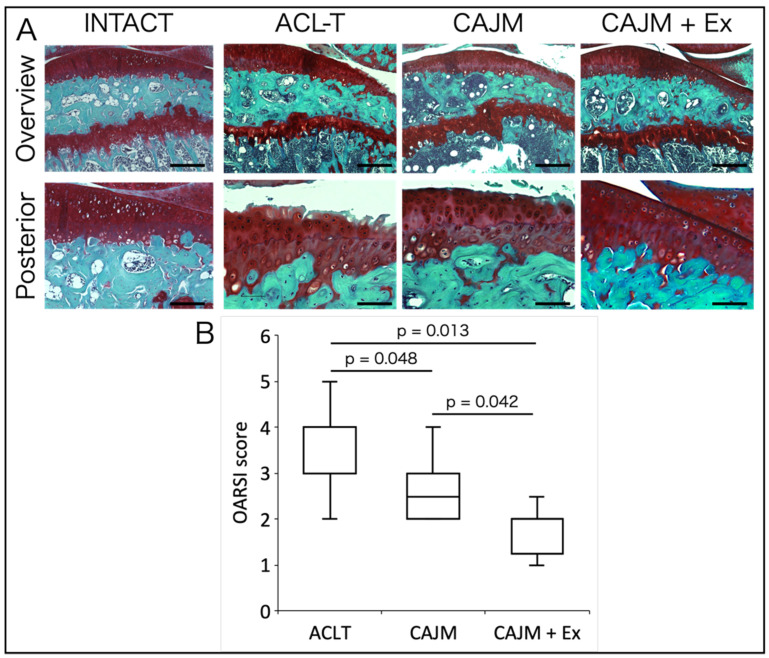
Results of Safranin-O Fast Green staining (**A**) and OARSI scoring of cartilage defects (**B**). The CAJM group had a lower OARSI score than the ACL-T group, and the CAJM + Ex group had even less cartilage degeneration (each group, *n* = 8). ACL-T, anterior cruciate ligament transection group; CAJM, controlled abnormal joint movement; Ex, treadmill exercise. Scale bar: Overview 200 μm, posterior 100 μm.

**Figure 4 life-11-00303-f004:**
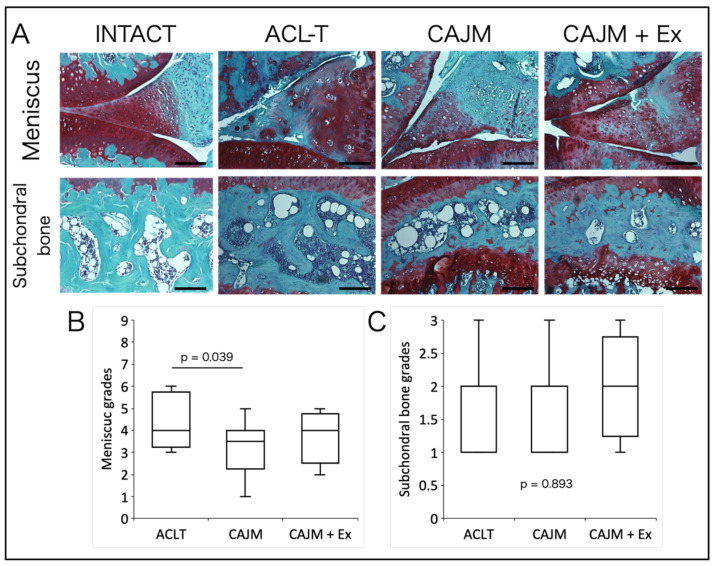
Histological images of the meniscus and subchondral bone with Safranin-O Fast Green staining (**A**). The CAJM group and the CAJM + Ex group had even less cartilage degeneration (**B**). Subchondral bone scoring results: no difference between the three groups (**C**) (each group, *n* = 8). ACL-T, anterior cruciate ligament transection group; CAJM, controlled abnormal joint movement; Ex, treadmill exercise. Scale bar: 100 μm.

**Figure 5 life-11-00303-f005:**
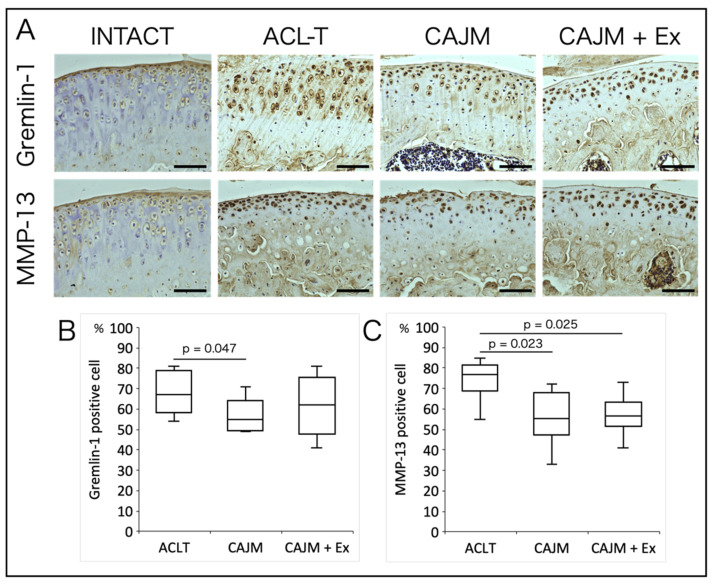
Histological images of immunostaining for articular cartilage (**A**). Gremlin-1 showed a decreased positive cell rate in the CAJM group (**B**), and MMP-13 showed a decreased positive cell rate in the CAJM and CAJM + Ex groups compared to that in the ACL-T group (**C**) (each group, *n* = 8). ACL-T, anterior cruciate ligament transection group; CAJM, controlled abnormal joint movement; Ex, treadmill exercise. Scale bar: 100 μm.

**Figure 6 life-11-00303-f006:**
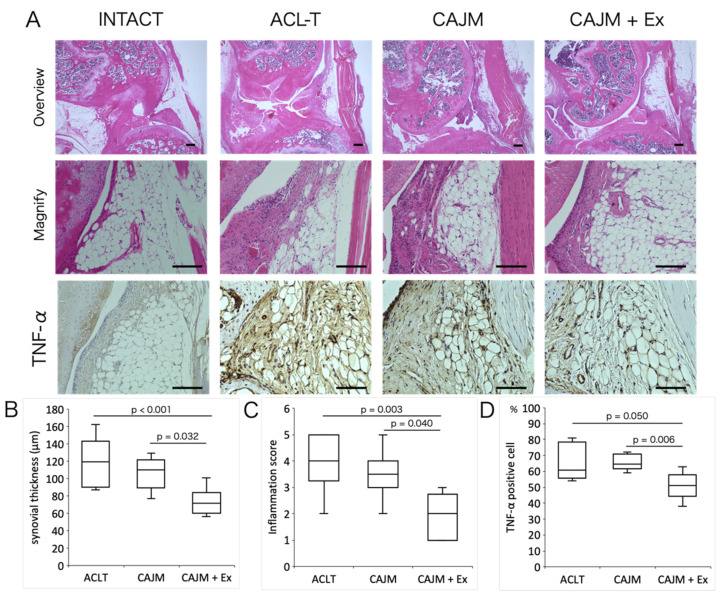
Histological images of synovium by HE staining, and histological images of immunostaining (**A**). The positive cell rate of TNF-α and the synovitis inflammation in the CAJM + Ex group was suppressed more than that in the ACL-T and CAJM groups (**B**–**D**) (each group, *n* = 8). ACL-T, anterior cruciate ligament transection group; CAJM, controlled abnormal joint movement; Ex, treadmill exercise. Scale bar: 100 μm.

## Data Availability

Data available in a publicly accessible repository.

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
