# Peer review of "Treadmill Exercise after Controlled Abnormal Joint Movement Inhibits Cartilage Degeneration and Synovitis"

_life, 2021, doi:10.3390/life11040303_

Round 1
Reviewer 1 Report
The study is interesting.
Abstract: Please state the number of animals.
Line 37-41: The authors state that “Currently, medication, physical therapy are the main conservative treatments for knee OA. However, although these therapies have been reported to be effective for improving pain, which is the main complaint of knee OA, the establishment of disease-modifying therapies that inhibit cartilage degeneration has not been realized.”.
It is unclear what the authors mean by medication and physical therapy. In my opinion, low-level laser therapy (LLLT) is worth mentioning as there is some evidence that LLLT with intensities of < 1000 mW/cm2 can improve cartilage in vivo [1]. This is plausibly due to its anti-inflammatory effects [2-6].
In the figures, please state what the error bars are (confidence intervals, standard deviation, standard error, e.g.).
- Xiang, A., et al., Laser photobiomodulation for cartilage defect in animal models of knee osteoarthritis: A systematic review and meta-analysis. Lasers in Medical Science, 2019.
- Tomazoni, S.S., et al., Isolated and combined effects of photobiomodulation therapy, topical nonsteroidal anti-inflammatory drugs, and physical activity in the treatment of osteoarthritis induced by papain. Journal of biomedical optics, 2016. 21(10).
- Tomazoni, S.S., et al., Effects of photobiomodulation therapy, pharmacological therapy, and physical exercise as single and/or combined treatment on the inflammatory response induced by experimental osteoarthritis. Lasers in Medical Science, 2017. 32: p. 101-108.
- Assis, L., et al., Aerobic exercise training and low-level laser therapy modulate inflammatory response and degenerative process in an experimental model of knee osteoarthritis in rats. Osteoarthritis and Cartilage, 2016. 24(1): p. 169-177.
- Pallotta, R.C., et al., Infrared (810-nm) low-level laser therapy on rat experimental knee inflammation. Lasers Med Sci, 2012. 27(1): p. 71-8.
- Santos, A., et al., Effects of low-level laser therapy on cartilage repair in an experimental model of osteoarthritis. Photonics & Lasers in Medicine, 2014. 3.
Author Response
I would like to thank the editors and reviewers for insightful comments. This has helped to greatly improve the manuscript. Following their comments, the manuscript was revised and changes in the document were identified with a red font. The point-by-point answers to these comments are shown below and are positioned at the end of the manuscript.
Comment #1
Please state the number of animals.
Response
We wish to thank the reviewer for pointing this out. I have stated in the number of animals in abstract as follows:
Lines 17–21
Twelve-week-old ICR mice (n = 24) underwent anterior cruciate ligament transection (ACL-T) surgery on their right knees and were divided into three groups as follows: ACL-T, animals in the walking group subjected to ACL-T; controlled abnormal joint movement (CAJM), and CAJM with exercise (CAJM+Ex) (n = 8/group).
Comment #2
Line 37-41: The authors state that “Currently, medication, physical therapy are the main conservative treatments for knee OA. However, although these therapies have been reported to be effective for improving pain, which is the main complaint of knee OA, the establishment of disease-modifying therapies that inhibit cartilage degeneration has not been realized.”.
It is unclear what the authors mean by medication and physical therapy. In my opinion, low-level laser therapy (LLLT) is worth mentioning as there is some evidence that LLLT with intensities of < 1000 mW/cm2 can improve cartilage in vivo [1]. This is plausibly due to its anti-inflammatory effects [2-6].
Response
We wish to thank the reviewer for this comment. As the reviewer pointed out, physiotherapy, such as LLLT, has also been reported to be effective for cartilage protection. Therefore, we added physiotherapy to the previous list of medication and physical therapy for the treatment for OA.
Lines38-39
Currently, medication, physiotherapy, such as LLLT [6–10], and physical therapy are the main conservative treatments for knee OA.
Comment #3
In the figures, please state what the error bars are (confidence intervals, standard deviation, standard error, e.g.).
Response
We wish to thank the reviewer for pointing this out. Parametric data are shown as the mean with 95% confidence interval, whereas non-parametric data are shown as the median with interquartile range.
This is described in the Statistical analysis section.
Lines 180–182
Parametric data are shown as mean values [95% confidence intervals], and non-parametric are shown as median values [interquartile range].
Reviewer 2 Report
The work by Oka et al, describes changes within the joint environment due to movement restriction after induction of OA by ACLT. The work is complementary to the work published by the same group in Cartilage in 2019, by adding Gremlin-1 staining, synovitis, and meniscus score. Importantly, there are some serious methodological issues in the present study:
-All findings are based on histological analysis. However, the scoring is conduced by only one investigator. To prevent bias, the scoring should be done by at least two independent scorers who are blinded for the treatment. Additionally, it is even more important to describe the scoring methods in more detail. How many sections were used per knee, were scores averaged? Etc etc.
-The scoring system used to evaluate meniscus damage is based on the score by Pauli et al. However, this score is validated for human menisci. Scoring of mice menisci is more appropriate by the scoring system described by Kwok et al, 2016 Osteoarthritis Cartilage, especially since degeneration based on SafO staining is inverted in human and mice.
-The evaluation of synovial inflammation is not done appropriate as the authors refer to a cartilage scoring system by Prizker. Scoring of synovial inflammation should be done either by measuring synovial thickness on the histological slides or by a scoring system such as described by Jackson et al 2014 Arthritis Rheumatology. It s also not mentioned where the synovial inflammation in evaluated, in the medial/lateral side, or the parapatellar groove?
Minor points:
-consider to change “controlled abnormal movement” to “joint movement restriction”.
-the number of mice per group should be mentioned in the method text, not only in the legend of Fig 1. The methods section would benefit if the ARRIVE guidelines for reporting animal studies is followed.
-The panels with the histological pictures should be larger and should include scale bars
Author Response
I would like to thank the editors and reviewers for insightful comments. This has helped to greatly improve the manuscript. Following their comments, the manuscript was revised and changes in the document were identified with a red font. The point-by-point answers to these comments are shown below and are positioned at the end of the manuscript.
Comment #1
All findings are based on histological analysis. However, the scoring is conduced by only one investigator. To prevent bias, the scoring should be done by at least two independent scorers who are blinded for the treatment. Additionally, it is even more important to describe the scoring methods in more detail. How many sections were used per knee, were scores averaged? Etc etc.
Response
We wish to express our deep appreciation to the reviewer for their insightful comments. Recently, there were some papers that used only one researcher’s evaluation to conduct histological analysis, and thus, this study was also conducted by only one researcher.
However, to prevent bias, it is better for two or more researchers to perform the evaluation, and thus, we modified the evaluation method. I asked two independent evaluators to perform the evaluation and treated the average value as the representative value. In addition, we evaluated two sections with 100 μm intervals.
Lines 150–153
Two independent observers performed scoring, and the average value was retained. For the evaluation of cartilage, meniscus, subchondral bone, and synovitis, two sections were evaluated each. The two sections were spaced 100 μm apart.
Comment #2
-The scoring system used to evaluate meniscus damage is based on the score by Pauli et al. However, this score is validated for human menisci. Scoring of mice menisci is more appropriate by the scoring system described by Kwok et al, 2016 Osteoarthritis Cartilage, especially since degeneration based on SafO staining is inverted in human and mice.
Response
We wish to express our deep appreciation to the reviewer for their insightful comments. Indeed, Pauli et al. is a report in which the evaluation of human meniscus was performed. Kwok et al.’s report is a study on mice, and this paper is more applicable to our evaluation, and thus, we modified the evaluation method.
Lines142-145
The posterior meniscus was evaluated with reference to the report by Kwok et al. [26]. The total scores for all criteria (structural, cellularity, and matrix staining) ranged from 0 to 21.
Comment #3
-The evaluation of synovial inflammation is not done appropriate as the authors refer to a cartilage scoring system by Prizker. Scoring of synovial inflammation should be done either by measuring synovial thickness on the histological slides or by a scoring system such as described by Jackson et al 2014 Arthritis Rheumatology. It s also not mentioned where the synovial inflammation in evaluated, in the medial/lateral side, or the parapatellar groove?
Response
We wish to thank the reviewer for this comment. My analysis might have lacked an assessment of synovitis. Therefore, as the reviewer pointed out, I have added the measurement of synovial thickness. The results of the analysis showed that synovial thickness was suppressed in the CAJM + Ex group, similar to the scoring results.
Lines 146–150
HE (hematoxylin–eosin) staining was performed to evaluate the synovitis of the patellar groove. The evaluation of synovitis was performed based on the OARSI score [28] and by measuring synovial thickness. Synovial thickness was quantified using dedicated image analysis software (Image J; National Institutes of Health, Bethesda, MD, USA).
Comment #4
-consider to change “controlled abnormal movement” to “joint movement restriction”.
Response
We wish to thank the reviewer for this suggestion. Our model tries to restore the abnormal joint movement caused by ACL-T, not “restrict” the joint motion. Therefore, we would like to retain the phrase “controlled abnormal movement”.
Comment #5
-the number of mice per group should be mentioned in the method text, not only in the legend of Fig 1. The methods section would benefit if the ARRIVE guidelines for reporting animal studies is followed.
Response
We wish to thank the reviewer for pointing this out. The numbers for each group are now listed in the methods section.
Lines 91–94
Twelve-week-old male ICR mice (n = 24) were used as subjects. The mice were divided into three groups (n = 8/group) as follows: ACL-T group (untreated after ACL-T), con-trolled abnormal joint movement (CAJM) group (joint braking only), and CAJM+Ex group (joint braking followed by treadmill exercise).
Comment #6
-The panels with the histological pictures should be larger and should include scale bars
Response
We wish to thank the reviewer for pointing this out. We apologize for the small figure, and this has been fixed. The scale bar contains the 95% confidence interval.
Reviewer 3 Report
Oka et al decribe the effect of controlled abnormal joint movement and treadmill exercise on the progression of surgically induced osteoarthitits (OA). They use an ACL transection model and study OA progression using histological and immunohistological stainings. In general, the topic is intersting as there is a urgent need for novel therapeutic approaches in OA treatment.
My major concern with this study is that important controls are lacking to fully support the conclusions. Even though the main goal was to study the effect of CAJM and EX an ACL-T, age-matched untreated controls, sham operated animals and CAJM+Ex without CAJM would be relevant.
Almost all figures are much too small to see any details. There is enough space to present the data in larger images and higher magnifications. The number of animals analyzed with a specific method should be given in the figure legend.
The analysis of the subchondroal bone is rather superficial. I can understand that not all methodological approaches like eg µCT, are available. However, on the images presented in Figure 4A it looks like the BV/TV is increased in CAJM+EX. How representative are the pictures that are shown?
It should be explained ith one or two sentences why greminl-1 and MMP-13 were stained. In addition, the authors should discuss why an intracellular staining (and the percentage of positive cells) of an extracellularly active MMP is relevant. Why is there no extracellular staining for MMP-13 at the site of action?
How do the autors know, that exercise has an anti-inflammatory effect (as mentioned in the introduction)? Is this based in the observation that synovitis is reduced with less TNFa expressing cells?
Minor comments:
The introduction could be better structured and provide some more information about CAJM and what is known about benficial effects of exercise.
In the abstract, the authors should clearly state in which direction the effects wnet and not only write that it was different.
Line 39: ‚reducing pain‘ sounds better than ‚improving pain‘.
In the discussion, the authors mention a number of cell types (like M1-macrophages) and factors that could be studied in ICH and/or analysed in serum samples. I would recommend to extend the analysis and better exploit the model in the future.
I think that the text would benefit from revision by an native speaker. In addition, there are a lot of typos that should be corrected.
Author Response
Reviewer 3
Please see attached.

Round 2
Reviewer 2 Report
The manuscript by Oka et al, has improved over the previous version by utilizing suitable scoring methods such as mice meniscus score by Kwok et al and measurement of synovial thickness. Moreover, the authors have provided more clear histological pictures in order for proper interpretation of the results. Yet, there are still concerns, mainly due to the lack of details on methodology as also mentioned in the first review round. Important details regarding the histological methods are still missing which is important since the entire manuscript is based on histological data and are needed for the reader for interpretation and be convinced of the results.
- As mentioned in the previous review round, the authors score synovitis based on the OARSI score by Pritzker et al (ref 28), yet this reference only describes a score for structural cartilage damage. It is therefore unclear on what criteria the synovitis score is based. The scoring tables/criteria should be provided as supplementary data.
- As mentioned in the previous round, the ARRIVE guidelines for reporting animal experiments should be followed. Reporting these details on animal supplier, housing conditions, number of animals in one cage all help the reader to interpret the results better since the model is based on joint movement.
- How are the knees sectioned, sagittal, coronally? In which quadrant was the cartilage scoring done?
- Istotype control staining must be provided for all immunohistological stainings.
- More details regarding the measurement of synovitis on histological slides should be mentioned. How many locations per knee were measured? Unilateral or bilateral? These details should all be mentioned.
- Fig 6: it is still unclear what the location is of the pictures shown here. The authors mention the patellar groove, yet there seems to be a lot of adipose tissue which may be remnants of the infrapatellar fat pad. The authors should provide an overview picture followed by the high magnification picture. If the authors had mentioned if the section is sagittal or coronal, this was also more clear. Related to that, the adipocytes appear to have increased TNFa staining. The authors should discuss these Results and discuss the implications.
Minor:
- Figures 2 and 3 are scrambled with overlapping images.
- Figure 3: why is the higher magnification picture labelled as posterior? The overview picture is also posterior then?
- Line 482: the authors should refrain from identifying macrophages solely based on TNFa staining. Macrophages should be identified by a proper pan macrophage marker such as F4/80 or CD64.
Author Response
I would like to thank the editors and reviewers for insightful comments. This has helped to greatly improve the manuscript. Following their comments, the manuscript was revised and changes in the document were identified with a red font. The point-by-point answers to these comments are shown below and are positioned at the end of the manuscript.

Reviewer 3 Report
The authors addressed all my concnerns and commenst appropriately. Congratulations!
Author Response
We wish to thank the reviewer for comments.